# Social interactions drive efficient foraging and income equality in groups of fish

Roy Harpaz[1,2]*, Elad Schneidman[1]*

[1]Department of Neurobiology, Weizmann Institute of Science, Rehovot, Israel;
[2]Department of Molecular and Cellular Biology, Harvard University, Cambridge MA,
United States

**Abstract** The social interactions underlying group foraging and their benefits have been mostly studied using mechanistic models replicating qualitative features of group behavior, and focused on a single resource or a few clustered ones. Here, we tracked groups of freely foraging adult zebrafish with spatially dispersed food items and found that fish perform stereotypical maneuvers when consuming food, which attract neighboring fish. We then present a mathematical model, based on *inferred* functional interactions between fish, which accurately describes individual and group foraging of real fish. We show that these interactions allow fish to combine individual and social information to achieve near-optimal foraging efficiency and promote income equality within groups. We further show that the interactions that would maximize efficiency in these social foraging models depend on group size, but not on food distribution, and hypothesize that fish may adaptively pick the subgroup of neighbors they 'listen to' to determine their own behavior.

**\*For correspondence:**
roy_harpaz@fas.harvard.edu (RH);
elad.schneidman@weizmann.ac.il
(ES)

**Competing interests:** The authors declare that no competing interests exist.

## Introduction

Living in a group has clear benefits, including expansion of sensory sensitivity (*Miller and Bassler, 2001*; *Pratt et al., 2002*; *Berdahl et al., 2013*; *Ward et al., 2011*), sharing of responsibilities and resources (*Clutton-Brock et al., 1999*; *McGowan and Woolfenden, 1989*; *Chittka and Muller, 2009*), collective computation (*Miller and Bassler, 2001*; *Pratt et al., 2002*; *Berdahl et al., 2013*; *Ward et al., 2011*; *Karpas et al., 2017*; *Shklarsh et al., 2011*; *Ward et al., 2012*), and the potential for symbiotic relations between members that would allow for specialization by individual members (*Chittka and Muller, 2009*; *Gordon, 1996*). Understanding the interactions among individuals that give rise to macroscopic behavior of groups is therefore central to the study and analysis of collective behavior in animal groups and other biological systems.

Group foraging is a prominent example of collective behavior, and social interactions among members have been suggested to increase foraging efficiency (*Pitcher, 1986*; *Radakov, 1973*; *Giraldeau and Caraco, 2000*; *Zahavi, 1971*; *Brown, 1988*; *Pitcher et al., 1982*) by allowing individuals to combine private and social information about the location of resources and their quality (see however (*Laland and Williams, 1998*; *Giraldeau et al., 2002*) for adverse effects of social information). Theoretical models have been used to study social foraging using various strategies, including producer-scrounger dynamics (*Barnard and Sibly, 1981*) and the use of 'public' information (*Clark and Mangel, 1984*; *Bhattacharya and Vicsek, 2014*; *Valone, 1989*). These models explored the efficiency of the underlying strategies and their evolutionary stability, as well as the effects of group composition and the distribution of resources (*Barnard and Sibly, 1981*; *Bhattacharya and Vicsek, 2014*; *Tania et al., 2012*; *Torney et al., 2011*; *Clark and Mangel, 1986*; *Hein et al., 2016*). Experimental work, in the field and in the lab, aimed to identify interactions between foraging individuals (*Ward et al., 2012*; *Coolen et al., 2001*; *Harel et al., 2017*; *Krebs, 1974*; *Flemming et al., 1992*; *Miller et al., 2013*) and their dependence on factors such as the distribution of resources, phenotypic diversity among foragers, animal personality, and foraging strategies of mixed species

(*Ryer and Olla, 1995*; *Farine et al., 2014*; *Michelena et al., 2009*; *Aplin et al., 2014*; *Jolles et al., 2017*). The interaction rules studied with these theoretical models and the emerging group behavior had mostly qualitative correspondence to that of real groups, as the predictions of theoretical models were usually not tested against experimental data of groups at the individual level (*Barnard and Sibly, 1981*; *Martínez-García et al., 2013*; *Hamilton and Dill, 2003*; *Mottley and Giraldeau, 2000*; *Giraldeau and Beauchamp, 1999*).

Moreover, most models studying how schooling or flocking may improve foraging efficiency have explored the case of single sources, or clumped food patches (*Karpas et al., 2017*; *Shklarsh et al., 2011*; *Ward et al., 2012*; *Hein et al., 2016*; *Miller et al., 2013*; *Aplin et al., 2014*; *Jolles et al., 2017*). Yet, in many real-world situations, animals are likely to encounter distributed food sources, where maintaining a tight group may not be beneficial for all group members. Indeed, schooling and shoaling species have been shown to disperse when confronted with distributed resources (*Ryer and Olla, 1995*; *Miller and Gerlai, 2007*). A characterization of group foraging for multiple sources with complex spatial distribution is therefore needed.

The ability to track the behavior of groups of individuals with high temporal and spatial resolution under naturalistic conditions (*Pérez-Escudero et al., 2014*; *Ballerini et al., 2008*; *Greenwald et al., 2015*; *Nagy et al., 2010*; *Dell et al., 2014*) makes it possible to explore foraging models and behavior quantitatively in groups and individuals. Here, we studied food foraging by groups of adult zebrafish in a large arena, where multiple food items were scattered and individuals could seek them freely. We inferred social interaction rules between fish and compared the accuracy of several mathematical models of group foraging based on these rules and the swimming statistics of individual animals. We explored the performance of these models in terms of foraging dynamics and efficiency of food consumption by the group and of individuals for various resource distributions and group sizes. Our data-driven approach allowed us to analyze foraging strategies through the local dependencies between conspecifics and to show that social interactions increase foraging efficiency in real groups of fish. We further used these models to study the effect of social interactions on income equality between members of the group. Finally, we used our models to explore the implications of social interactions on the efficiency of collective foraging of larger groups under different distributions of resources, and asked how animals could choose an optimal foraging strategy under varying conditions.

## Results

We studied free foraging of single adult zebrafish and of groups of three or six fish in a large circular arena with shallow water, where small food flakes were scattered on the surface (*Figure 1*, *Figure 1—figure supplement 1A*, and Materials and methods; All data used in this study is avilabale online). The trajectories and heading of individual fish, the position of flakes, and food consumption events were extracted from video recordings of these experiments using a tracking software that was written in house (*Harpaz et al., 2017*). Tracking of fish identities in the videos were verified and corrected using *IdTracker* (*Pérez-Escudero et al., 2014*). Fish were highly engaged in the foraging task and consumed all flakes in less than 2 min in most cases (*Figure 1B*, *Figure 1—figure supplement 1C*, *Figure 1—video 1*). The number of identified consumption events varied between groups, since in some cases fish ate only a part of a flake or flakes disintegrated into smaller parts (*Figure 1—figure supplement 1B*). The time differences between flake consumption events reflect the nature of foraging and its efficiency. We found that in our setup, the time it took a group of $k$ fish to consume $n$ flakes, $T_k(n)$, was accurately predicted using a simple exponential model,

$$log\ \hat{T}_k(n) = \frac{n}{b_k} + a_k \tag{1}$$

where parameters $b_k$ (the 'consumption rate' of the fish) and $a_k$ (time to detection of the first flake) were found numerically to minimize the mean squared error between the model predictions and the data. The correlation between the observed $T_k(n)$ and model predictions $\hat{T}_k(n)$ was very high, with median $r^2$ values: 0.94 [0.09], 0.96 [0.04], 0.96 [0.05] for groups of one, three, and six fish, respectively (brackets show interquartile range). Larger groups consumed the flakes faster than smaller ones (*Figure 1B-C*), and the feeding rates were higher than those predicted just from having a larger number of foragers, which we tested using a model of independent foragers (*Figure 1—figure*

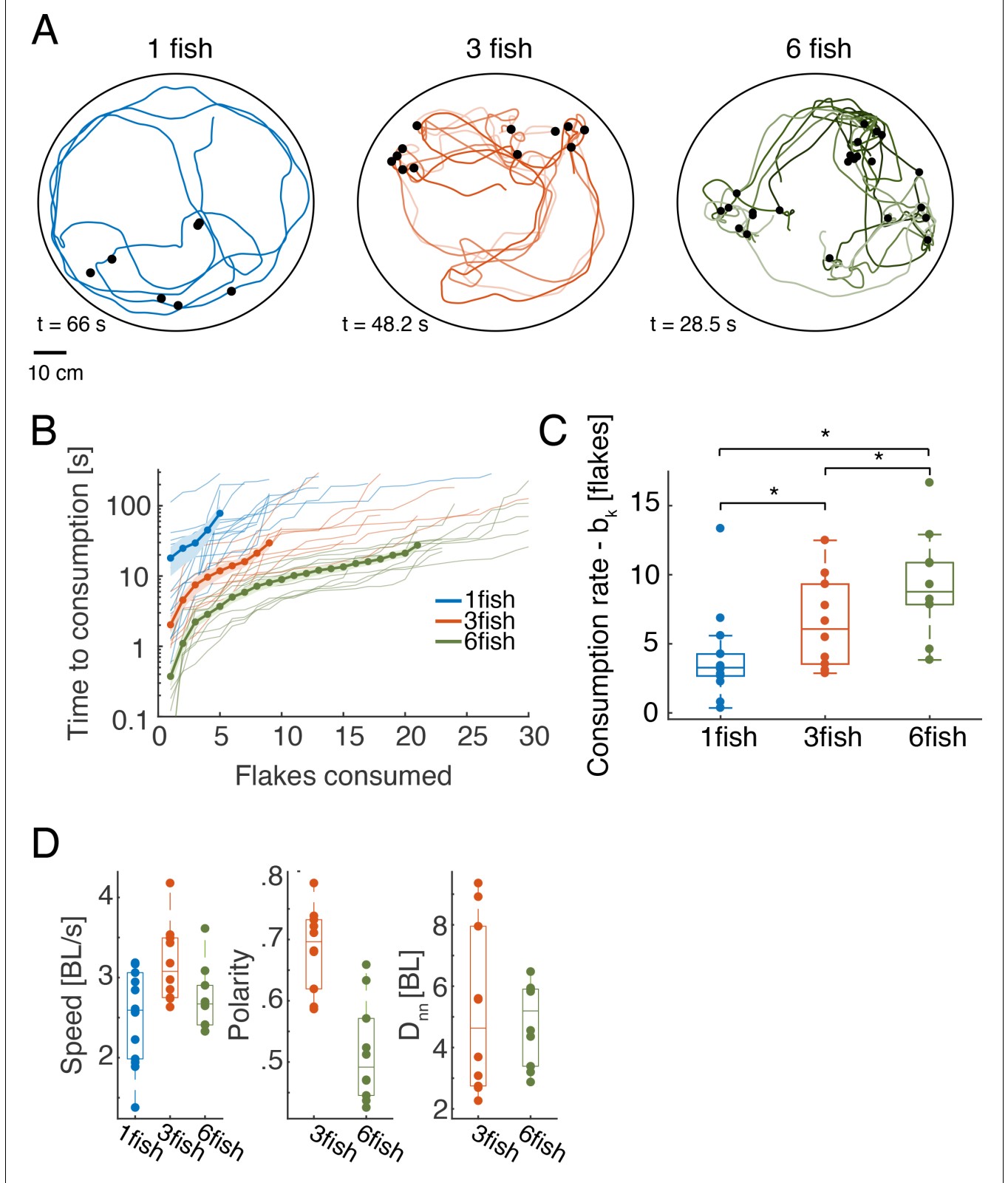

**Figure 1.** Individual and group foraging by adult zebrafish. (**A**) Example trajectories of groups of one, three, and six fish foraging for food flakes. Colored lines show the trajectories of individual fish, black dots show the location of flakes consumed by the fish. (**B**) Time of consumption of the i[th] flake in the arena is shown for each of the groups tested (thin lines show N = 14, 10, and 10 groups of one, three, and six fish, correspondingly) overlaid with the mean of all groups for every group size (thick line); light shaded areas represent SEM. (Because individual groups of the same size did not

*Figure 1 continued on next page*

*Figure 1 continued*

always consume the same total number of flakes, averages were calculated over the first 5, 9, and 21 flakes consumed by the groups of one, three, and six fish, respectively, and the number of consumed flakes is truncated at 30 for clarity; the full curves are shown in *Figure 1—figure supplement 1C*; We emphasize that all analyses were conducted on the full curves). (C) Boxplots show the rate of flake consumption $b_k$ that was fitted for each of the groups shown in B. Middle horizontal lines represent median values and box edges are the 1$^{st}$ and 3$^{rd}$ quartiles; asterisks denote p<0.05 under Wilcoxon's rank-sum test, N = 14,10,10 groups of fish. (D) Average speed, polarity, and nearest neighbor distance for individual fish and for the groups. Horizontal lines represent median values and box edges are the 1$^{st}$ and 3$^{rd}$ quartiles.

The online version of this article includes the following video and figure supplement(s) for figure 1:

**Figure supplement 1.** Experimental design for studying free foraging behavior in zebrafish groups.
**Figure 1—video 1.** Foraging behavior of fish in a group.
https://elifesciences.org/articles/56196#fig1video1

*supplement 1D*, see models below). Fish in groups of different size differed also in their average swimming statistics, with groups of three fish exhibiting higher swimming speeds, higher polarization, and larger variation of group cohesion (nearest neighbor distances) (*Figure 1D*). We also found that group polarity was highly correlated with group cohesion for both groups of three and six fish, but was not correlated with swimming speed (*Figure 1—figure supplement 1E*). We therefore asked what social interactions between fish may underlie group foraging and swimming trajectories.

## Characterizing social interactions during foraging

To explore the nature of interactions between foraging fish, we analyzed individual swimming behavior before and after flake consumption events and found that fish performed salient and stereotypic maneuvers around the consumption of food items. They increased their speed when approaching food and then abruptly turned in the process of consuming it (*Figure 2A–C* and *Figure 2—video 1*). This maneuver was characterized by a decrease in speed and an increase in the curvature of the trajectory (*Figure 2B–C*, *Figure 2—figure supplement 1A*). We found that for most groups of 3 and 6 fish analyzed (75%), neighboring fish responded to these salient behaviors and were more likely to visit areas of recent flake consumption within 1–4 s (*Figure 2D–E*), much more than expected by chance (p<0.05 for 3 and 6 fish, Wilcoxon's signed rank test). Fish were less attracted to the location of a neighbor's consumption maneuver if flakes were more abundant in the arena (*Figure 2—figure supplement 1B*). To confirm that these salient behavioral changes attracted fish to previous consumption areas and not a physical trace of the flake, we also analyzed 'pseudo consumption events', where fish performed speed changes similar to those seen near flake consumption, but with no food present (see Materials and methods). Neighboring fish were attracted to such pseudo-consumption events, affirming that fish responded to the specific behavior of their neighbors (*Figure 2—figure supplement 1C–D*).

## Social interaction models of group foraging

To study the implications of attraction to locations of feeding by other fish, we simulated foraging groups of fish with various social interactions and without them (see Materials and methods). Simulations were based on the swimming characteristics of real fish and the empirical spatial distributions of flakes (*Figure 1A*, *Figure 3A-B*). The swimming trajectory of each fish was simulated by successive drawing from the distribution of step sizes (the length of the path traveled on discrete 'bouts' according to our segmentation of real fish trajectories; *Figure 3A, B*) and turning angles (change of heading angle between two discrete bouts) of a specific fish in the real group (*Figure 3A-B*, *Figure 3—figure supplement 1A-B*). However, if a flake was within the 'range of detection' by a fish ($D_f$), then that fish oriented itself directly towards the flake with a probability that monotonically decreased with the distance to the flake (see *Figure 3C*). The independent foraging model (IND) is based on a collection of such fish. In addition, we considered six social interaction models that combine attraction to neighbors' feeding events and attraction and alignment between fish regardless of feeding (Attraction to feeding events- Att$_{feed}$; Attraction to neighbors- Att; Alignment with neighbors- Align; and their combinations: Att$_{feed}$+Align, Att+Align, Att$_{feed}$+Att+Align). In all these models, the direction of motion of each fish was modulated by the behavior of neighboring fish within the 'neighbor detection range' $D_n$ (*Figure 3D* and Materials and methods) (*Bhattacharya and*

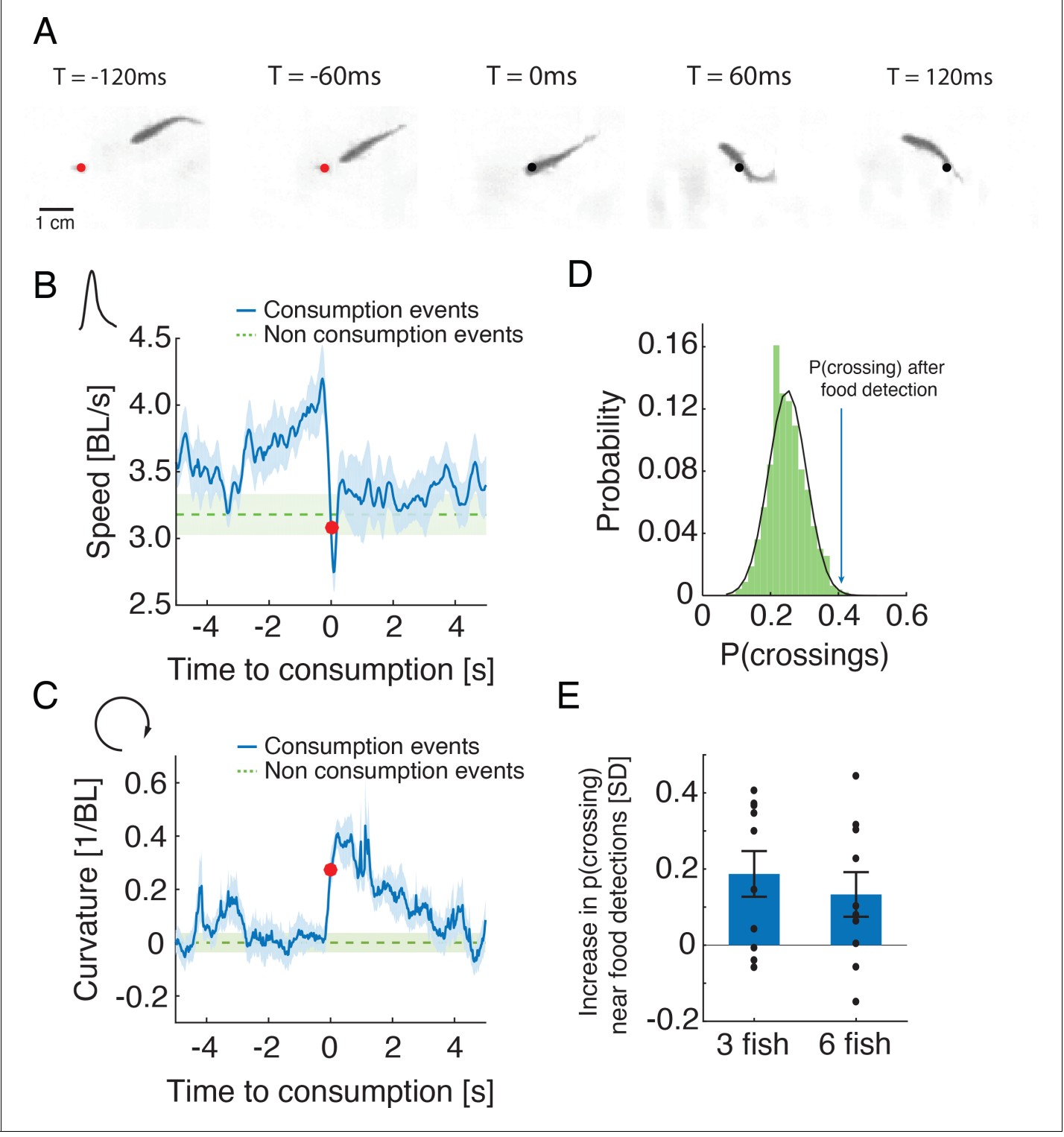

**Figure 2.** Stereotypical maneuvers before and during flake consumption by one fish attracts its neighbors. (A) An example of the stereotypical behavior of one fish (in a group of three) showing flake detection and consumption. Flake position is shown by a red circle before consumption and by a black circle after it has been eaten. (B–C) Stereotypical behaviors around flake consumption (denoted by time 0 and marked by a red dot) include a transient increase in speed (shown in body length (BL) per second), followed by a sharp decrease (in B); this is accompanied by an increase in the curvature of the trajectory (in C). Bold blue lines are mean speed and curvature profiles over all detection events of groups of three fish, and dotted green lines show a reference value calculated from random points along the trajectories not related to consumption events (curvature values were normalized such that the average curvature is zero). Light blue and green shadings show SEM. (D) Probability of neighbors to cross within 2 BLs from

*Figure 2 continued on next page*

*Figure 2 continued*

the location of a previous flake consumption, within 1–4 s following a consumption event, for one group of three fish (blue arrow), compared to the distribution of such neighbor crossing events, if flake consumption events were ignored (Materials and methods). This reference distribution of crossings is well fitted by a Gaussian distribution (mean = 0.25, SD = 0.056), which is shown by a overlaid black line. (E) Crossing probabilities for groups of three and six fish show significant increase from the baseline neighbor crossing distribution of each group, similar to C; 0 represent the mean of the baseline crossing distributions, error bars represent SEM.

The online version of this article includes the following video and figure supplement(s) for figure 2:

**Figure supplement 1.** Stereotypical maneuvers before and during flake consumption by one fish attracts its neighbors.

**Figure 2—video 1.** Stereotypical maneuvers of fish during flake detection and consumption.

https://elifesciences.org/articles/56196#fig2video1

---

*Vicsek, 2014*; *Torney et al., 2011*; *Couzin et al., 2002*; *Huth and Wissel, 1992*; *Vicsek et al., 1995*). These different models allowed us to tease apart the specific contribution of attraction to neighbor's previous flake consumptions and of the general schooling or shoaling tendencies of the fish (see *Figure 3—figure supplement 1C* and Materials and methods for a description of all models ; all simulation codes are avilable online).

For each of the real groups in our data, we simulated each of the models using a range of possible values of the model's parameters $D_f$ and $D_n$ (*Figure 3—figure supplement 1C*, Materials and methods). For each set of $D_f, D_n$ values, we estimated the accuracy of the models in predicting the sequence of consumption events as well as two swimming statistics of the group: the average polarity of the group (or alignment between fish) and the average cohesion of the group (average distance to the nearest neighbor -$D_{nn}$) (*Figure 4A-B*, Materials and methods). We evaluated the performance of each model on each of these measures (*Figure 4C*), and their combination (*Figure 4D*). We found that the IND model did not describe well the consumption times of the groups or their swimming statistics (*Figure 4C*). Simple attraction to neighbors (Att model) also failed to accurately represent the consumption times or the polarity of the group, yet it accurately described distances between fish and slightly improved overall accuracy over the IND model (*Figure 4C-D*). Alignment interactions among fish (Align model) was significantly better than the Att model, specifically in describing the polarity of the groups and the sequence of consumptions, indicating that fish respond to their neighbors' direction while foraging (P<0.005, for groups of three fish with N=10, and P<0.01 for groups of six fish with N=10; Wilcoxon's signed rank test)(*Figure 4C-D*, *Figure 4—figure supplement 1A*). The most accurate model of fish swimming and foraging behavior was the one that included both alignment and attraction to neighbors' previous consumption events (Att$_{feed}$+Align), showing high accuracy in describing both the swimming statistics of the groups and the sequence of consumptions, which was significantly better than the competing models (P<0.005 when comparing the Att$_{feed}$+Align to all other models shown, N=10,10 for groups of three and six fish, Wilcoxon's signed rank test)(*Figure 4C-D*, *Figure 4—figure supplement 1A*). The inferred range of social interactions of the best fitted model (Att$_{feed}$+Align) were ~4 and 8 times larger than the range of flake detection (median $D_n$ values: 21.5 [11], 11.5 [6] and median $D_f$ values: 2.5[2], 3[3] for groups of three and six fish, interquartile range in parenthesis; see *Figure 4—figure supplement 1C*).

Importantly, the observed improvement in accuracy was not a result of increased model complexity: First, the number of model parameters was the same for all social models. Second, models that include attraction to neighbors regardless of flake consumption (Att+Align, Att$_{feed}$+Att+Align models) were less accurate than the Att$_{feed}$+Align model (*Figure 4—figure supplement 1B,D*). We conclude that while fish continuously respond to the swimming direction of their neighbors, they also exhibit a specific attraction to neighbors' previous flake consumptions during foraging.

## Increased foraging efficiency is predicted by attraction to neighbor's flake consumptions

To understand the impact of social interactions on foraging efficiency, we compared the feeding rates of foraging fish in the best fit social model (Att$_{feed}$+Align model) and the reference IND model (*Figure 4*), for a wide range of model parameter values $D_f$, $D_n$ (*Figure 5A*). The feeding rates were accurately approximated by an exponential function (as in Eq. 1; $R^2$>0.98 for all simulations). As

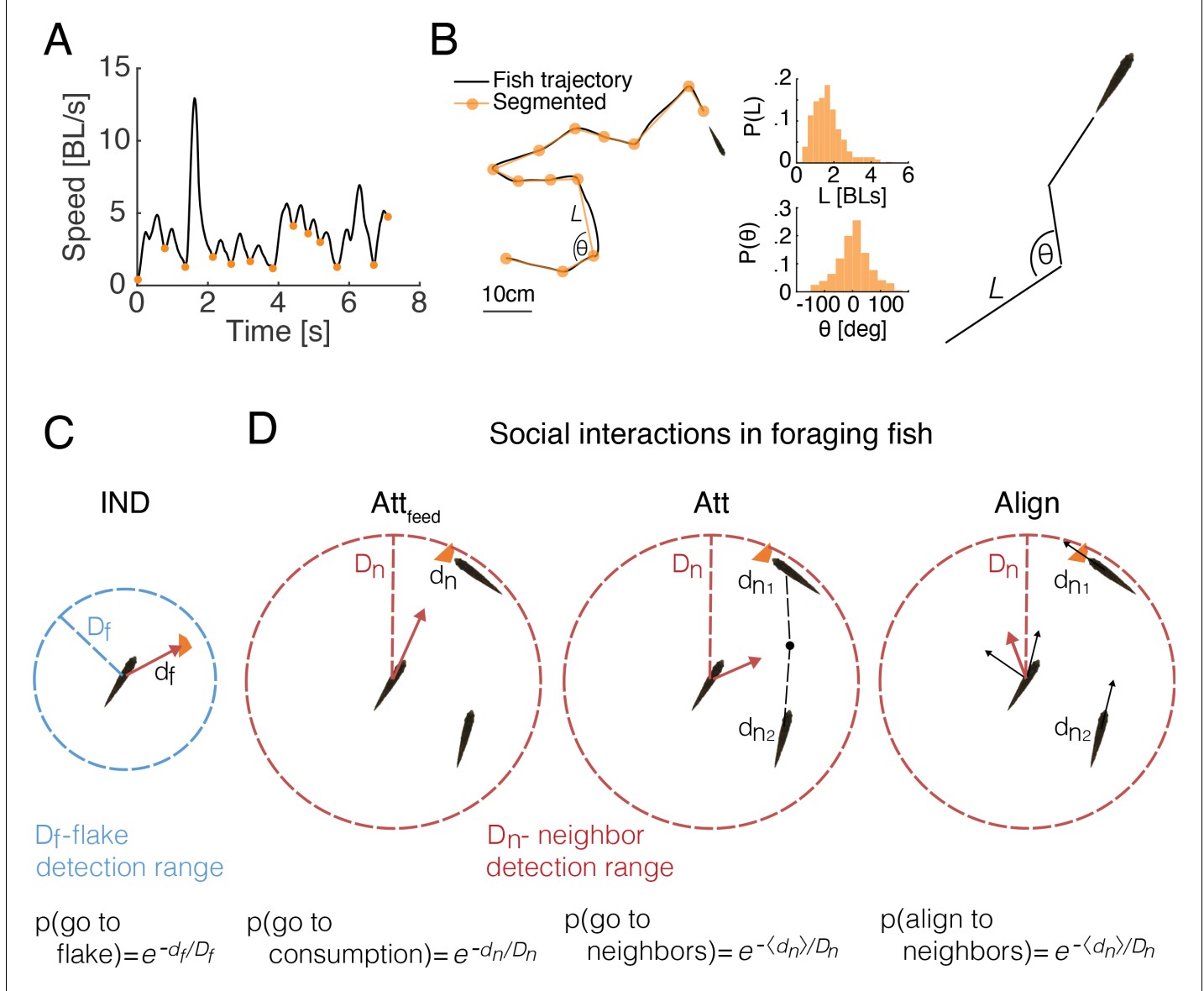

**Figure 3.** Comparing social and independent foraging using model simulations. (**A**) An example of the speed profile of an individual fish in a group. Orange dots mark local minima and are used to segment the continuous motion into discrete events. (**B**) Left: A snippet of a fish trajectory, corresponding to the speed profile in A and its segmentation into discrete steps (orange line). Middle: Distributions of step size $L$ and angle change $\theta$, between discrete steps over three fish in one of the groups (Materials and methods). Right: sketch of a simulated fish trajectory composed of successive drawings of $\theta$ and $L$ from the empirical distributions. (**C**) A sketch of the independent model of fish foraging: At each time step, if there was a flake within a fish's detection range ($d_f < D_f$ depicted by the blue circle), the fish oriented towards the flake with a probability p(go to flake). (**D**) Sketches of the different social interactions between fish. Each fish may detect consumption of flakes by another fish (left), if that fish was within the neighbor detection range ($d_n < D_n$ red circle). The observing fish was then attracted to the consumption point with probability p(go to consumption). Additionally, a fish may respond to the swimming behavior of its neighbors within $D_n$, regardless of flake consumption, by swimming towards the average position of the neighbors (middle) with probability p(go to neighbors) or by aligning its swimming direction (right) with the neighbors that are within $D_n$, p(align to neighbors). Different combinations of these possible social interactions comprise the six different social models that we tested (**Figure 3—figure supplement 1C**, Materials and methods).

The online version of this article includes the following figure supplement(s) for figure 3:

**Figure supplement 1.** Models of fish foraging behavior.

expected, the simulated groups consumed flakes faster as $D_f$ increased (**Figure 5B**). For relatively short flake detection range ($D_f \leq 6$ BL), flake consumption rates increased with $D_n$, reflecting the

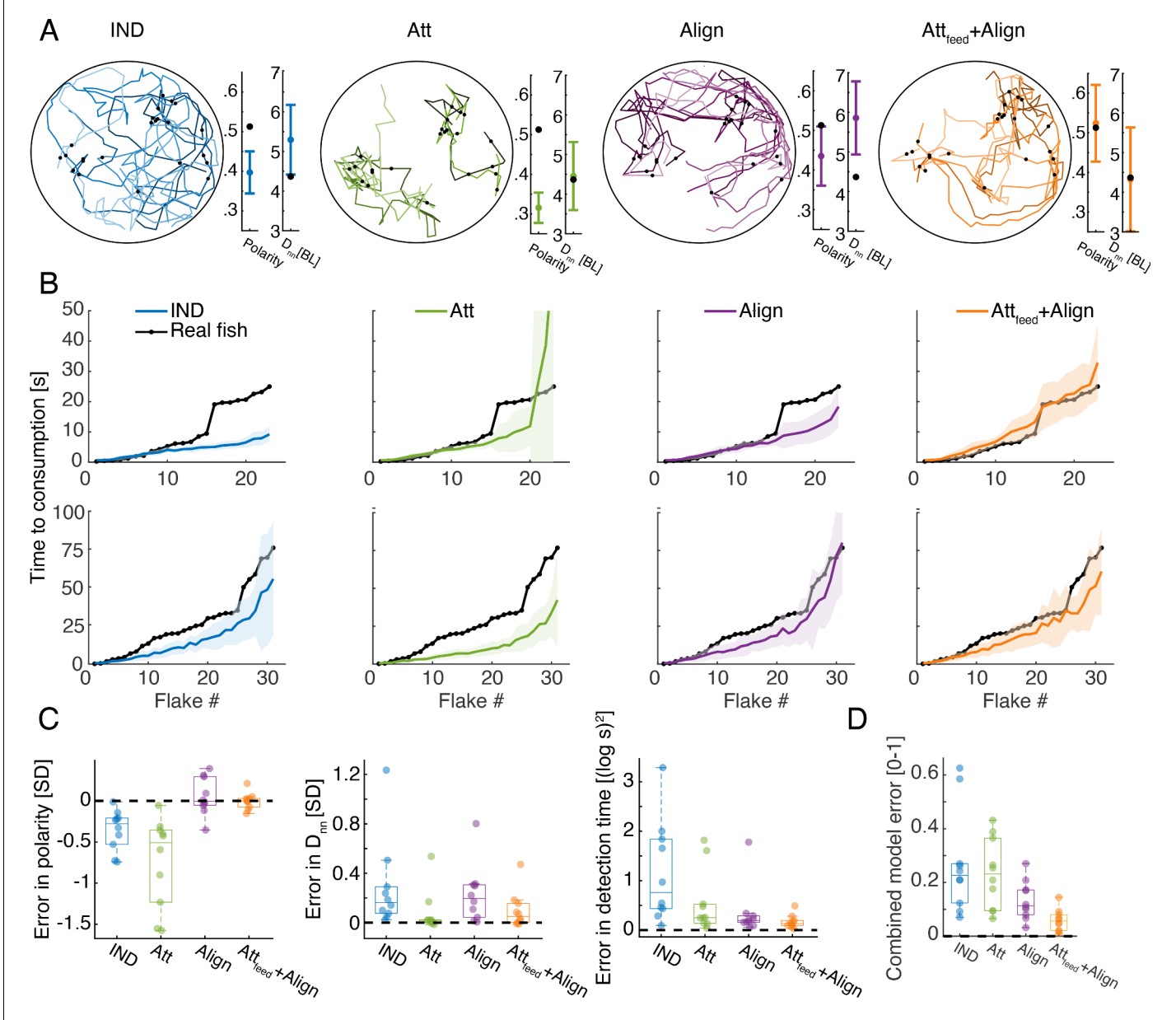

**Figure 4.** Social models incorporating attraction to neighbors' flake consumptions give the best fit to real foraging groups. (**A**) Example trajectories from simulations of foraging of a group of six fish, for the IND, Att, Align, and the Att$_{feed}$+Align models that use the parameters that gave the best fit to real group foraging. Colored lines show different individual fish and black dots are flake positions. Next to the simulated trajectories, we plot the average group polarity and nearest neighbor distance in the simulations (colored dots), and the experimental values of the real foraging group (black dots); Error bars represent standard deviation (SD) in the simulation. (**B**) Flake consumption times (black dots) of two groups of six fish (Top row shows the group whose trajectories are shown in A) and the average and standard deviation of the best-fit models (bold colored lines represent averages; shaded areas represent SD). (**C**) Errors of best fit models for groups of six fish are shown for three statistics of interest: the polarity of the group $E^{polarity} = \frac{P^{data} - P^{model}}{SD(P^{data})}$, the nearest neighbor distance $E^{D_{nn}} = \frac{D_{nn}^{data} - D_{nn}^{model}}{SD(D_{nn}^{data})}$, and the consumption times $E^{consumptions} = \frac{1}{N} \sum_n \left[ log(t^{data}(n)) - log(t^{model}(n)) \right]^2$, where N is the number of flakes consumed. Dots represent different experimental groups; horizontal lines are median values and boxes represent the 1$^{st}$ and 3$^{rd}$ quartiles. Dotted line represents 0 error in prediction or a perfect fit to the data. (**D**) Combined error of each of the models presented in C. $E^{combined} = \left( E^{polarity} + E^{D_{nn}} + E^{consumptions} \right)/3$, where all error measures were scaled to be between 0 and 1, by dividing by the largest observed error for that measure. The Att$_{feed}$+Align model is significantly more accurate than the IND, Att, and Align models (p<0.005 for all, Wilcoxson's signed rank test, N = 10 groups, Materials and methods). Each dot represents one group; horizontal lines are median values and boxes represent the 1$^{st}$ and 3$^{rd}$ quartiles.

The online version of this article includes the following figure supplement(s) for figure 4:

*Figure 4 continued on next page*

*Figure 4 continued*

**Figure supplement 1.** Social models incorporating attraction to flake consumption by neighbors show the best fit to real foraging groups.

effect of directly responding to neighbors' foraging behavior. For $D_f > 6$ BL, increasing $D_n$ had a very little effect on consumption rates (*Figure 5B*).

Social interactions in the Att$_{feed}$+Align model resulted in a significant increase in consumption rates, compared to the IND model, only for simulations with low $D_f$ and high $D_n$ values (red areas in *Figure 5C*). Importantly, most groups of real fish were best matched by simulated groups with parameter values that were well within the area of the parameter space were social interactions improve foraging efficiency (low $D_f$ and high $D_n$), approaching the peak of the expected improvement in foraging performance (*Figure 5C*). The observed improvement due to social interactions was model specific - social interaction models that did not include attraction to neighbors' previous flake consumptions (e.g. the Align, Att, or Att+Align models) did not show a similar improvement over the independent model (*Figure 5D*, *Figure 5—figure supplement 1A-B*). Moreover, social foraging strategies that included attraction to neighbors' positions (not specifically related to flake consumptions) were less efficient than independent foragers (*Figure 5D*, *Figure 5—figure supplement 1A-B*).

## Individual efficiency and income equality in socially interacting fish

We next explored how the Att$_{feed}$+Align foraging strategy might affect the foraging success of individual members of the group. We simulated groups in which only a fraction of the foragers used social interactions, while the others foraged independently (*Figure 6A* illustrates these mixed strategy groups). Comparing foraging success of the social and non-social individuals within the same group, we found that individuals using social information consumed up to 20% more flakes than their non-social companions, and this advantage decreased as the number of interacting agents in the group increased (*Figure 6B-C*, *Figure 6—figure supplement 1A*). These effects were most pronounced in models that used the same parameter range that matched real foraging groups, namely low $D_f$ and high $D_n$ (*Figure 6—figure supplement 1A*).

We further assessed the equality of food distribution among individuals in real groups and simulated groups using the Theil index of inequality (*Theil, 1967*):

$$I_{Theil}(k, n) = \frac{1}{k} \sum_{i=1}^{k} \frac{n_i}{\mu} log\left(\frac{n_i}{\mu}\right),$$                    (2)

where $k$ is the number of fish in the group, $n_i$ is the number of flakes consumed by the $i^{th}$ fish, and $\mu = \frac{n}{k}$ is the average number of flakes consumed by a fish in the group. We normalized $I_{Theil}$ by its maximal possible value, $log\, k$ (the case where one fish consumes all flakes), and measured equality using $1 - \frac{I_{Theil}}{log\, k}$, which ranges between 0 (a single fish who consumed all flakes) and 1 (full equality). Real groups showed high equality values, with median values of 0.89 [0.21] for groups of three fish and 0.92 [0.16] for groups of six fish (values in brackets show the interquartile range). The corresponding simulated groups using the Att$_{feed}$+Align strategy exhibited similar high equality values (Median = 0.92 [0.05], 0.90 [0.1] for simulated groups of three and six fish), indicating that resources were distributed relatively equally among real and simulated fish (*Figure 6—figure supplement 2A*). We then compared income equality for the different social foraging models and found that only models that include specific attraction to neighbors' previous flake consumptions exhibited increased equality compared to independent foraging (*Figure 6—figure supplement 2C*). In simulated groups using the Att$_{feed}$+Align strategy, equality was a function of both the number of socially interacting individuals within the group and the spatial dispersion of the flakes (*Figure 6—figure supplement 2B*). For low spatial dispersion of flakes, simulated groups composed of just independent foragers showed the highest inequality of all groups with mixed strategies, and equality increased with the number of foragers in the group that used the Att$_{feed}$+Align strategy (*Figure 6D-E*, *Figure 6—figure supplement 2DA*). When all individuals in the group used the Att$_{feed}$+Align strategy, equality was higher than in groups of independent foragers for groups of three and six fish, for most parameter values that match real foraging groups (*Figure 6D-E*). In contrast, in

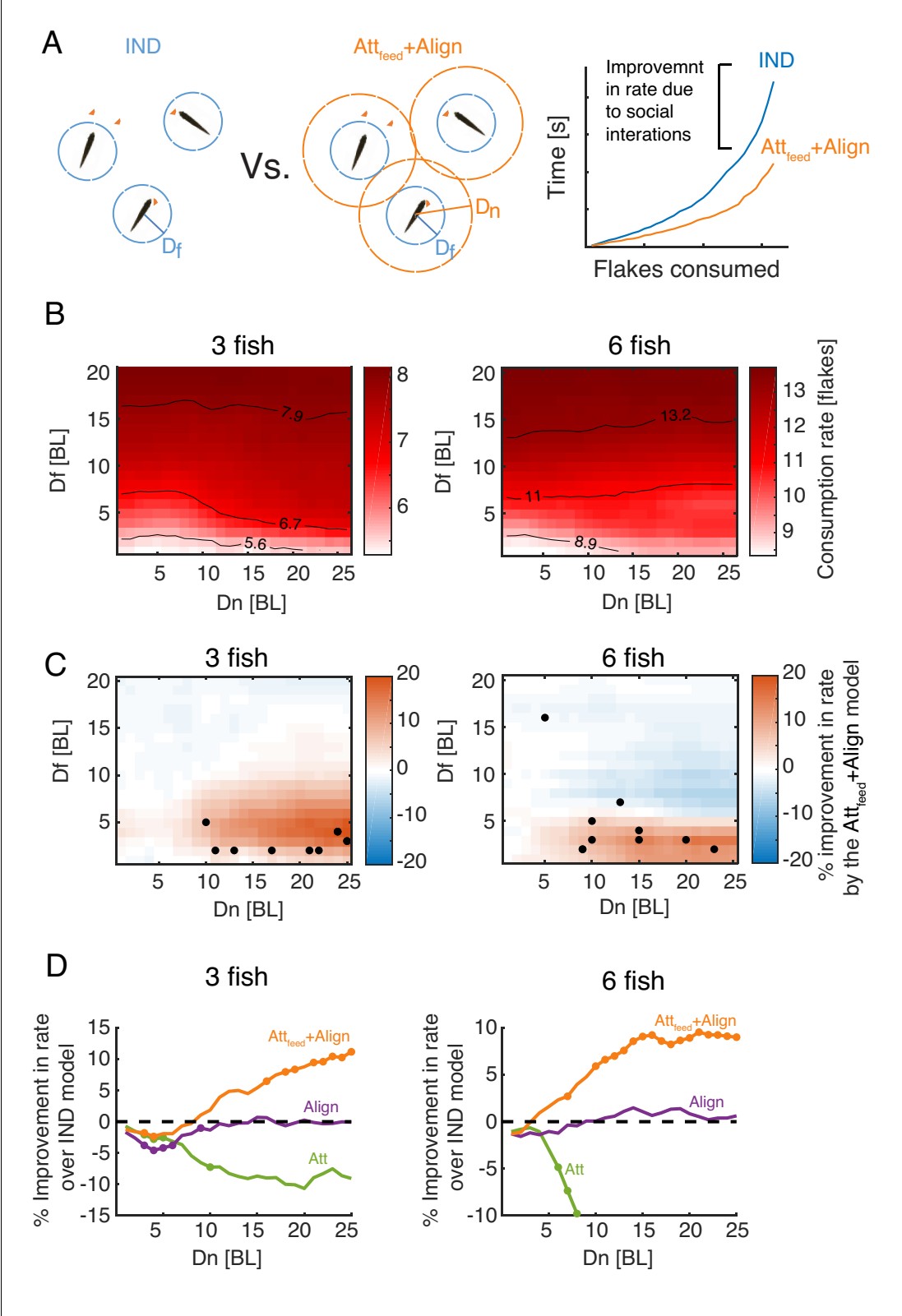

**Figure 5.** Attraction to neighbors' feeding results in increased foraging efficiency. (**A**) *Left*: Sketch of two groups of three fish foraging, with their different interaction ranges $D_f$, $D_n$ overlaid. For $D_n = 0$, the group is composed of independent foragers (IND model). *Right*: foraging efficiency was estimated by comparing the slope (*b*; see eq. 1) of the exponential function fitted to the rate of flake consumption of socially interacting agents (Att_feed+Align model) and independent (IND) foragers. (**B**) Average consumption rates, *b*, for different combinations of $D_f$ and $D_n$; the first column on

*Figure 5 continued on next page*

Figure 5 continued

the left ($D_n = 0$) represent independent foragers. Contours denote 10, 50, and 90% of the highest observed rate. (C) Difference in foraging efficiency for groups that utilize social interactions (Att$_{feed}$+Align) compared to groups of independent foragers (IND) for all model parameters. Dots represent $D_f$ and $D_n$ values of simulated groups that best fitted real foraging groups. (D) Average improvement in the rate of flake consumption by socially interacting individuals compared to independent foragers. Colors indicate different social foraging strategies; dotted black line represent no change compared to independent foragers (IND); results were averaged over all simulations with $D_f \leq 5$ which was the parameter range where real groups were matched by simulations. Colored dots represent statistically significant differences (P<0.05, Wilcoxon's signed rank test).

The online version of this article includes the following figure supplement(s) for figure 5:

**Figure supplement 1.** Foraging efficiency and attraction to neighbors' consumption events.

environments with high dispersion of flakes these effects disappeared or even reversed, with groups of three fish showing only a moderate non-significant increase in equality and groups of six fish showing a significant decrease in equality (*Figure 6D-E*). Thus, attraction to neighbors' consumption events increases income equality among group members only when resources are hard to come by.

## Simulating larger groups and different distributions of food

Finally, we investigated the predictions of the Att$_{feed}$+Align foraging model for larger groups and additional spatial distributions of food. We simulated groups of up to 24 fish in environments with spatial distributions of food ranging from a single cluster of food items to a uniform distribution (*Figure 7A*, *Figure 7—videos 1–6*). The increase in efficiency due to social interactions was most pronounced when food items were highly clustered in space, whereas for the extreme cases of random or uniform distributions, the models predict that social interactions would hinder foraging performance (*Figure 7B*, *Figure 7—figure supplement 1A*; *Giraldeau and Caraco, 2000*; *Ryer and Olla, 1995*; *Ryer and Olla, 1998*; *Ranta et al., 1993*). Importantly, our simulations predict that groups of 12 and 24 fish that follow the Att$_{feed}$+Align strategy would be less efficient than independent foragers for almost all the food distributions we tested. This is mainly due to the fact that the larger groups are more cohesive and disperse less in the environment, making the search less efficient (*Figure 7—video 6*). A social foraging strategy that only includes attraction to neighbors' flake consumption events (without a tendency to align with neighbors - Att$_{feed}$ model) increased foraging efficiency also for the large group sizes (*Figure 7—figure supplement 2A*). Income equality in the group also increased in clustered food distributions and decreased in non-clustered ones, for all group sizes (*Figure 7C*, *Figure 7—figure supplement 1B*, *Figure 7—figure supplement 2B*).

Interestingly, simulated groups of three and six fish were most efficient for intermediate $D_n$ values in the cases of clustered and real flake distributions (~10-12 BL, see *Figure 7B*). Simulations of groups with longer social interaction ranges added only a small gain to foraging efficiency. In contrast, for simulated groups foraging with non-clustered food distributions (Random and Grid, *Figure 7A*), increasing $D_n$ values resulted in decreased efficiency for larger groups, but had almost no effect for groups of three fish. These simulations suggest that regardless of the flake distribution, optimal interaction ranges for groups of three fish could be long, while groups of six fish should use intermediate interaction ranges to balance their gains at high clustered environments with their losses at distributed environments. The parameter values of the best fit models to real groups conform with these predictions with median $D_n$ values of 21.5 and 11.5 for groups of three and six fish, respectively.

## Discussion

We studied free foraging behavior in groups of adult zebrafish and found that fish responded to the salient swimming maneuvers of shoal mates that indicated the consumption of food, by swimming to these locations. Mathematical models of group behavior that combined the tendency of fish to align with one another and to attract to the locations of previous flake consumptions by other fish, accurately described fish foraging behavior and their success rates, and were superior to several other (commonly used) social interaction models. This foraging strategy increased efficiency of groups specifically in models that best matched real foraging groups, improved income equality within the groups, and was efficient under different resource distribution settings. Simulations of the models also show that socially interacting individuals that would rely on attraction to feeding events

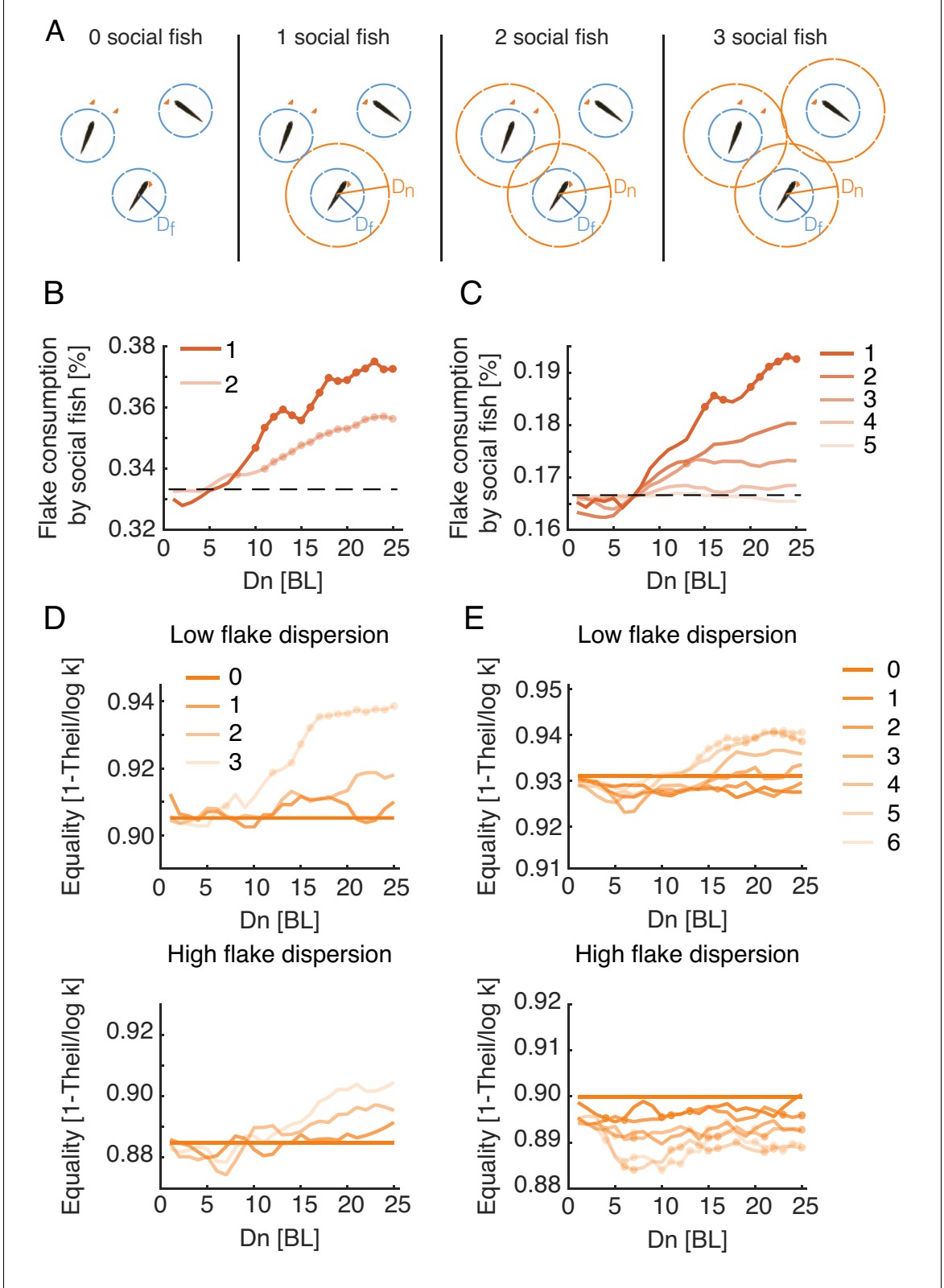

**Figure 6.** Social interactions promote individual foraging efficiency and income equality within groups. (A) Sketch showing simulated groups of three fish with a varying number of foragers who use the Att$_{feed}$+Align strategy during foraging. (B, C) Average fraction of flakes consumed by the socially interacting individuals in the same simulated group. Colors indicate the number of social foragers in the group; dotted lines show the expected consumption values if resources were distributed equally among foragers; results were averaged over all simulations with $D_f \leq 5$, which was the

*Figure 6 continued on next page*

*Figure 6 continued*

parameter range where real groups were matched by simulations; dots represent significant increase in efficiency of social foragers over independent foragers within the group (Wilcoxon's signed rank test, N=10, 10). (D, E) Equality of food distribution within groups, measured by $1 - \frac{I_{Theil}}{\log k}$ (Eq. 2). Lighter colors represent a larger fraction of social foragers. Data is shown for both low flake dispersion environments (top), and high dispersion (bottom) (*Figure 6—figure supplement 2B*). Equality values were averaged over $D_f \leq 5$; dots show a significant difference from independent foragers (0 social fish) (Wilcoxon's signed rank test; N=5 for each group size and dispersion level).

The online version of this article includes the following figure supplement(s) for figure 6:

**Figure supplement 1.** Social interactions improve foraging efficiency of individuals in groups.
**Figure supplement 2.** Social interactions improve income equality in groups.

by other fish would consume more food than shoal mates that forage independently. Our results thus present a detailed social foraging heuristic that matches fish behavior in a naturalistic context, and constitutes a highly efficient and robust foraging strategy.

Our modeling predicts that the inferred interaction ranges that best fit real foraging groups would result in a robust foraging strategy for groups of three and six fish for various spatial distributions of food. This implies that to forage efficiently fish could adjust their interaction range according to the perceived group size, regardless of the (usually unknown) distribution of food. Additionally, the reduction in foraging efficiency predicted by our models for larger simulated groups (12 and 24 fish) predicts that these groups are more likely to break down into smaller groups that will exhibit increased efficiency. This finding is consistent with the observation that zebrafish both in the wild and in the laboratory are rarely found in cohesive groups of 12 fish or more (*Harpaz et al., 2017*; *Suriyampola et al., 2016*). Interestingly, when simulating groups that only utilize attraction to neighbors' consumption events (without the general schooling tendency observed in real fish) the models predicted increased efficiency for all group sizes. We therefore hypothesize that this interaction type represents a general behavioral strategy for individuals foraging in a social context, also for non-schooling species (*Brown, 1988*; *Clark and Mangel, 1984*; *Flemming et al., 1992*; *Miller et al., 2013*).

Since zebrafish rely heavily on their visual system (*Saverino and Gerlai, 2008*; *Pita et al., 2015*), our modeling focused on vision as the main source of social information between individuals. It is likely that other sensory modalities, namely tactile or odor pathways, also play a role in information transfer during foraging. However, the inferred parameters of the best fit models in our data indicated that neighbor detection ranges were ~4–8 times larger than flake detection ranges - reaching up to 21 body lengths, on average. It is unclear whether odors or tactile information may be detected from such large distances on such short time scales (*Moore and Crimaldi, 2004*). Thus, while various modalities may modulate fish behavior, we suggest that vision plays the prominent role in processing social information during foraging.

We focused here on attraction-based strategies for the fish, since in our experiments fish were attracted to areas where neighbors detected food. However, previous studies have suggested other search strategies that were based on repulsion between individuals (*Tani et al., 2014*) or on maximizing information about the location of food (*Karpas et al., 2017*). Although our modeling framework gave an excellent fit to the data, it is possible that foraging fish combine or alternate between strategies in different environmental conditions, based on group composition or their internal state (*Farine et al., 2014*; *Michelena et al., 2009*; *Harpaz et al., 2017*; *Kurvers et al., 2011*). Thus, models that incorporate additional strategies according to an explicit policy, might prove to be even more accurate in explaining fish behavior.

Observations of groups in nature, and related theoretical models, suggest that groups may contain a fraction of individuals whose search is based on their personal information ('producers') as well as individuals that rely mostly on social information ('scroungers') (*Barnard and Sibly, 1981*; *Mottley and Giraldeau, 2000*; *Harten et al., 2018*). Our results suggest that when individuals in the group have similar foraging capabilities and a limited social interaction range (*Bhattacharya and Vicsek, 2014*; *Bhattacharya and Vicsek, 2015*), using both individual and social information is the most efficient strategy for the individual. An interesting extension of our models would be to explore individual differences between members of the group and their effect on individual social foraging strategies (*Aplin et al., 2014*; *Jolles et al., 2017*; *Michelena et al., 2010*; *Brown and Irving, 2014*),

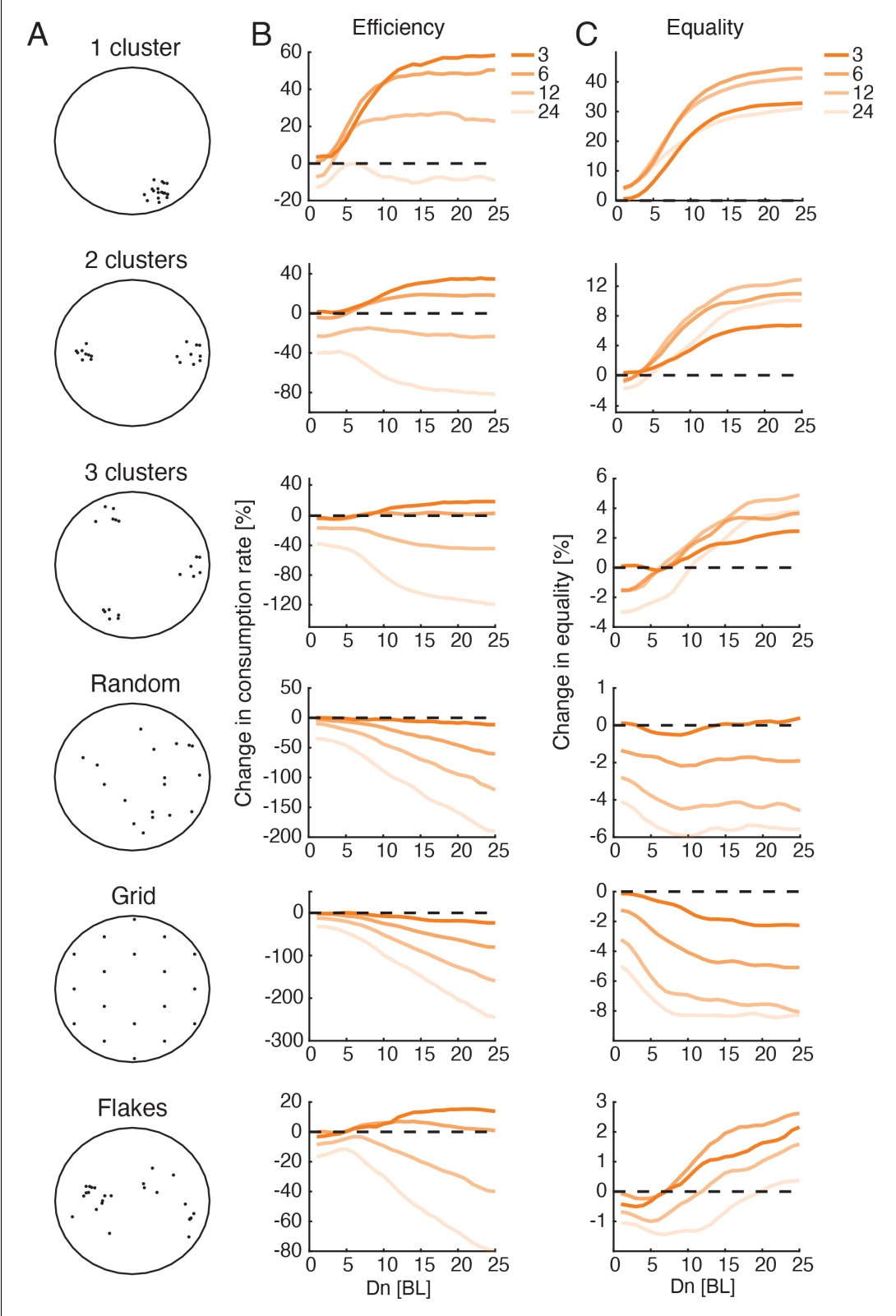

**Figure 7.** Optimal interaction range depends on group size and not on resource distribution. (**A**) Sketches showing six different resource distributions used for foraging simulations, three with varying levels of clustering (1, 2, and 3 clusters), a random distribution, and uniform distribution (approximated by a hexagonal grid), and an example of the distribution of flakes taken from one of the experiments. (**B**) Improvement in group foraging rates (mean time to consumption per flake) of the Att$_{feed}$+Align model compared to independent foragers (IND model) for groups of 3, 6, 12, and 24 fish (dark to

*Figure 7 continued on next page*

*Figure 7 continued*

light colors) for the different spatial distributions (panels from top to bottom). Results were averaged over all simulations with $D_f \leq 5$ (*Figure 7—figure supplement 1*, 100 repetitions per parameter combination, Materials and methods). (C) Same as in B but for the increase in equality of resources within groups.

The online version of this article includes the following video and figure supplement(s) for figure 7:

**Figure supplement 1.** Simulating different distributions of food.
**Figure supplement 2.** Optimal interaction range depends on group size and not on resource distribution.
**Figure 7—video 1.** Simulations of groups of six fish foraging in an arena with a single cluster of food flakes.
https://elifesciences.org/articles/56196#fig7video1
**Figure 7—video 2.** Simulations of groups of six fish foraging in an arena with 2 clusters of flakes.
https://elifesciences.org/articles/56196#fig7video2
**Figure 7—video 3.** Simulations of groups of six fish foraging in an arena with 3 clusters of flakes.
https://elifesciences.org/articles/56196#fig7video3
**Figure 7—video 4.** Simulations of groups of six fish foraging in an arena with a uniform distribution of flakes.
https://elifesciences.org/articles/56196#fig7video4
**Figure 7—video 5.** Simulations of groups of six fish foraging in an arena with random distribution of flakes.
https://elifesciences.org/articles/56196#fig7video5
**Figure 7—video 6.** Groups of 12 simulated fish foraging in an environment with 3 clusters of flakes.
https://elifesciences.org/articles/56196#fig7video6

or the existence of stable sub-groups of individuals with higher tendencies to interact with one another in larger groups of foragers.

Finally, we note that our work reflects the power of detailed behavioral analysis of individuals in real groups for building accurate mathematical models of social interactions. Learning the models from the data and testing them on real groups allowed us to explore the efficiency and robustness of the interactions among group members in a quantitative manner. This data-driven approach would be applicable and potentially imperative in the analysis of social behavior of large groups of individuals - where macroscopic or mean field like models would not suffice to characterize the interplay between complex and possibly idiosyncratic traits of individuals and emerging group behavior.

## Materials and methods

### Experimental setup

Detailed description of the setup is given in *Harpaz et al., 2017*. Briefly, adult fish were purchased from a local supplier (Aquazone LTD, Israel) and housed separately in their designated groups for more than a month prior to behavioral experiments. Fish were housed in a standard holding system consisting of a recirculating multistage filtration system where temperature, conductivity, PH, and light- dark cycle were monitored.

Imaging of fish foraging behavior was done using an industrial video recording system composed of a *Vieworks VC-2MC-M340* camera with an 8 mm lens, a *Karbon-CL* frame grabber, and a recording server. Camera was suspended 150 cm above the experimental tank. During experiments we changed the effective tank size by using arenas of different size (*Figure 1—figure supplement 1A* ; see also discussion below). All water conditions were similar between the holding tanks and the experimental tank.

### Fish acclimation and behavioral experiments

To facilitate food searching behavior and to reduce fish anxiety, the following acclimation procedure was followed: On day 1, all fish were transferred to the designated experimental tank (D = 95 cm; water depth of 5 cm) and were allowed to explore the tank for 5 min. On days 2–5, all groups and individual fish were transferred from their home tanks to test tanks of increasing size (*Figure 1—figure supplement 1A*) where 6, 12, or 18 flakes were randomly scattered over the water in the area outside the start box (for groups of sizes 1, 3, and 6 respectively, *Figure 1—figure supplement 1A*). The number of flakes used in the experiments for individual fish and for groups were chosen based on preliminary experiments, as the amount of food that would entice single fish to engage in the

task yet not to overcrowd the arena with flakes (especially for the larger groups). Fish were first placed in a small starting box (25 cm x 25 cm) that was inside a larger arena. The small box was raised after 5 min and the fish were allowed to forage and consume the flakes in the larger arena for an additional 5 min. Fish were then netted and returned to their home tanks, keeping their original groups. Over the 4 days of training, we increased the size of the arena from the small starting box itself (day 2) to circular ones with D = 47.5, 67.2, 95 cm on days 3–5 (*Figure 1—figure supplement 1A*). On days 6–7 fish were kept in their home tanks and were deprived of food. Foraging was then tested on day 8. During training, no food was administered to the fish outside of the experimental arena. In total, *n = 106* adult fish (3 months old or older) were used at approximately 1:1 male to female ratio. Overall we tested 16 single individuals, 10 groups of three fish (30 fish in total), and 10 groups of six fish (60 fish in total). Two single individual fish were excluded from analysis as they did not swim when transferred to the experimental tank.

## Data extraction

Recorded videos were analyzed off-line to extract the size, position, speed, and orientation of individual fish, and the position of food flakes. Positional data was then used to estimate fish trajectories using a designated tracking software. All image processing and tracking were done using Matlab with software written in house; the details of these procedures are described in *Harpaz et al., 2017*. Fish identities were further corrected using *IdTracker* (*Pérez-Escudero et al., 2014*). Fish trajectories were smoothed using a Savitzky-Golay filter (*Savitzky and Golay, 1964*) spanning 17 frames which constituted ~1/3 of a second (all videos were recorded at 50 fps). Fish positions were defined as the coordinates of the center of each fish, $\vec{x}_i(t)$, and fish velocity was calculated as: $\vec{v}_i(t) = \frac{\vec{x}_i(t+dt)-\vec{x}_i(t-dt)}{2dt}$, with dt = 1 video frame or 20 ms. Direction of motion of the fish was defined as: $\vec{d}_i(t) = \frac{\vec{v}_i(t)}{|\vec{v}_i(t)|}$ and the trajectory curvature at time $t$ was given by $c(t) = 1/R(t)$, where R is the radius of the circle that gave the best Euclidean fit to a trajectory segment of length 600 ms, centered on time $t$.

## Tracking flakes

Flakes' locations were tracked with the same software that was used for tracking the fish (*Harpaz et al., 2017*). Flakes that were larger than four pixels (a radius of about 1.15 mm) were reliably detected. Flakes typically disappeared when eaten, but when consumed flakes broke into pieces, new smaller flakes appeared. Consumption events were defined at times when a fish made contact with a flake and the flake disappeared from the camera's field of view. The resolution of our camera did not allow us to confirm whether the fish digested the flake entirely.

## Flake consumption events and pseudo consumption events

We estimated the probability of fish to cross near the location of a consumption event by another fish, $P(crossing)$, by counting all events where at least one fish passed within 2 BL of that location within 1-4 s after a neighbor's flake consumption, and dividing it by the total number of consumption events. Since zebrafish tend to swim in groups, regardless of the presence of food, we compared this number to the probability of one fish to cross near a neighbor's position within 1-4 s when no food was recently consumed by that neighbor (within the last 4 s) or would be consumed in the near future (within the next 4s). We estimated this $P_{null}(crossing)$ by drawing $k$ random fish positions (mimicking $k$ flake detection events) 10000 times, from times when no flake was detected for at least 8 s (*Figure 2D*). The tendency to attract to the location of flake consumption was then given by: $P(crossing) - \langle P_{null}(crossing) \rangle$, where angle brackets represent the average over random drawings for a given group.

We defined 'pseudo flake consumptions' at times when fish exhibited a speed profile that was similar to that of a fish during real consumption events, namely gradual increase in speed followed by a sudden sharp decrease back to baseline (*Figure 2B*). To detect such events, we convolved the speed profile of individual fish in the group at all times when no flakes were present near the fish (for at least 8 s) with the calculated average speed profile near all real flake consumption events of that group (*Figure 2B*) and obtained a correlation measure for each point in time. We then treated the top 2.5 percent of this distribution as pseudo consumption events. The average number of

these events was 12.6 ± 4 and 21.8 ± 6.6 for groups of three and six fish, respectively. We compared the probability of neighbors to cross near the locations of such events, $P(pseudo\ crossing)$ to $P_{null}(crossing)$. The tendency of neighboring fish to cross near pseudo consumption events was high, and was correlated with their tendency to cross near real flake consumption events over groups (*Figure 2—figure supplement 1C-D*).

## Segmentation of fish trajectories

Segmentation of trajectories into discrete steps was based on the detection of local minima in the speed profile of the fish (*Figure 3A-B*). We characterized discrete steps by the total distance traveled between two minima points, $L$, and the change in angle between successive steps, or turning angle, $\theta$. These distributions were estimated for each fish and used to simulate their swimming behavior (*Figure 3—figure supplement 1A-B*).

## Simulating swimming behavior

### Fish motion

We modeled fish swimming behavior as a correlated random walk, defined by the distribution of step sizes, $L$, and of the turning angles between consecutive steps $\theta$ (see above). Thus, at each time point, we drew for each simulated fish a step size and a turning angle, which determined the direction of motion and the length of the next step (*Figure 3B*, *Figure 3—figure supplement 1A-B*). Average step size estimated from fish data was 2.85 ± 1.7 and 3.1 ± 1.9 body lengths (BL) for groups of three and six fish, and turning angle distributions were nearly symmetric with an SD of 50° and of 46° for groups of three and six fish (*Figure 3—figure supplement 1A-B*). The initial conditions of each simulation, or the starting positions of individual fish, matched those of the real fish groups. All simulations were conducted within bounded arenas, identical to those used for testing real fish. If a simulation step was expected to end outside of the arena boundaries, that movement was discarded and a new movement was chosen that did not exceed the arena boundaries. To compare simulated foraging time (counted as discrete steps) to real foraging experiments (*Figure 4B*), we divided the length of each simulated step by the calculated average speed of real foraging fish.

### Simulated flakes

In all simulations, we used the true positions of flakes as extracted from fish foraging experiments. If a flake drifted during an experimental epoch (due to water turbulence), we only used its final position in the simulations since the speed and distance were negligible compared to the motion of the fish. Since real flakes sometimes disintegrated into smaller bits after a consumption event, we copied that in the simulations, that is if flake $i$ at position $\vec{x}_i$ has appeared after flake $j$ was (partially) consumed, so did the corresponding flakes in the simulation.

### Sensory range of flake detection

Each simulated fish had a circular range $D_f$, within which it could detect a flake, and orient towards it with probability p(go to flake) $= e^{-d_f/D_f}$, where $d_f$ is the distance to the flake (*Figure 3C*). If a fish oriented itself toward a flake, its next step size was drawn from the empirical distribution. If the fish reached the flake (or passed it) during this movement, that flake was considered to be consumed. If the simulated fish did not reach the flake, the procedure was repeated. In the IND foraging model, $k$ such fish were simultaneously simulated, independent of one another.

### Sensory range of neighbor detection

Since foraging fish in real groups were found to be attracted to areas of previous flake consumptions, and since zebrafish are known to exhibit schooling and shoaling tendencies, we allowed agents in our simulated social models to detect and respond to neighbors' swimming and foraging behavior within the sensory range of neighbor detection, $D_n$. Specifically, agents in the social foraging models could combine various types of social interactions (*Figure 3D*, *Figure 3—figure supplement 1C*, and see below):

1. Attraction to neighbors' previous flake consumptions ('Att_feed'): if a neighbor of fish i was within the sensory range of neighbor detection, $D_n$, and found a flake in the previous $\tau$ time

steps, then fish i oriented towards the position of the previous consumption with probability p (go to consumption) = $e^{-d_n/D_n}$ , where $d_n$ is the distance to the position where a flake was consumed by a neighbor. In case a movement toward a consumption position was successfully drawn, the step size was drawn from the empirical distribution.

2. Neighbor attraction ('Att'): if neighbor(s) were found within the sensory range of neighbor detection, $D_n$ in the previous times step, fish oriented towards the center of mass of these neighbors with probability P(attract) = $e^{-\langle d_{n_j}\rangle/D_n}$ where $\langle d_{n_j}\rangle$ is the distance to the center of mass of the neighbors, such that the new direction of fish i is $\vec{d}_i(t+1) = \dfrac{\langle x_j(t)\rangle - x_i(t)}{|\langle x_j(t)\rangle - x_i(t)|}$, where $\langle x_j(t)\rangle$ is the center of mass of neighbors within $D_n$ at time t.

3. Neighbor alignment ('Align'): if neighbor(s) were found within the sensory range of neighbor detection, $D_n$ in the previous times step, fish adopted the average direction of these neighbors with probability P(align) = $e^{-\langle d_{n_j}\rangle/D_n}$ where $\langle d_{n_j}\rangle$ is the distance to the center of mass of the neighbors, such that the new direction of fish i is $\vec{d}_i(t+1) = \dfrac{1}{J}\sum\limits_{j\in d_{ij}<D_n} \vec{d}_j(t)$ where J is the number of neighbors within $D_n$.

## Hierarchical nature of the social models

Simulated fish actions were given by the following hierarchy: If a flake was within the $D_f$ range of a fish, that fish would turn towards it with p(go to flake). If a flake was not detected (i.e. no flake was within $D_f$), and a neighbor consumed a flake at a distance smaller than $D_n$, then the fish would move towards that location with the appropriate probability (for the cases where the model included response to food consumption by neighbors). If neither a flake nor a neighbor consumption event were detected, but neighbors were within a distance $D_n$, the fish responded to the position/orientation of these neighbors (given that the model included response to neighbors' swimming). If no flakes or neighbors were detected, or if motion towards these areas was not successfully drawn, then the next direction of motion of the fish was randomly chosen from the empirical turning angle distribution.

## Mixed strategy groups

In simulations of groups with mixed individual strategies, fish that used social interactions followed the procedure described above and the rest used independent foraging (*Figure 6A*).

## Simulating larger groups of foragers for various flake distributions

We simulated group foraging for various group sizes: 3, 6, 12 and 24 fish in a range of flake distributions with a constant number of 18 flakes (*Figure 7A*):

1. Single cluster. Flakes were randomly distributed within a small circle with a diameter of 9.5 cm (within the 95 cm diameter arena). Circle center was situated at a distance of 20 cm from the arena walls.
2. Two clusters. Flakes were equally distributed between two circles as described for the single cluster. Circle centers were situated on the diameter of the large arena to ensure maximal distance between clusters.
3. Three clusters. Flakes were equally distributed between three clusters as described above. Circle centers were positioned on the vertices of an equilateral triangle to ensure maximal distance between clusters.
4. Random. To achieve a random, non-clustered distribution we positioned flakes randomly in the arena and assessed their clustering level based on the nearest neighbor distance of flakes, using the clustering coefficient given in *Clark and Evans, 1954*: $C = \dfrac{NN_1}{0.5\cdot\sqrt{\rho}}$, Where $NN_1$ is the average nearest neighbor distance of all flakes, and $\rho$ is the density of flakes. Values of $|0 - C| < 0.01$ were taken to show no spatial clustering and were chosen for simulations.
5. Uniform (Hexagonal grid). We positioned N+1 flakes in a hexagonal grid within the arena boundaries. We then removed the center flake to allow fish to start in the center position.
6. Empirical flake distributions. We used the empirical flake distribution used in experiments (see above).

In all simulations, fish started foraging in the center of the arena (see *Figure 7—videos 1–6*). We repeated these simulations 100 times for all $D_f$ and $D_n$ values, for all group sizes in all simulated flake distributions (1-5). For the empirical flake distribution (see 6 above) we repeated simulations 20 times for all $D_f$ and $D_n$ values, for all group sizes and for every empirical flake distribution used in the experiments. We later averaged simulation results over the different flake distributions for each group size to achieve a single representative performance map similar to the results of the simulated flake distributions (*Figure 7—figure supplement 1*).

## Model classes and parameters

|  | Independent models | Social models |
|---|---|---|
| Df – flake detection range | 1–20 body length, increments of 1 | 1–20 body length increments of 1 |
| Dn – neighbor detection range | 0 body length | 1–25 body length increments of 1 |
| $\tau$ – memory of a neighbor detecting flakes |  | five time steps (where applicable) |
| Arena Radii | 32 BL (95 cm) | 32 BL (95 cm) |
| Turning angles | Estimated from swimming data of each fish |  |
| Step sizes | Estimated from swimming data of each fish |  |
| Flake positions | According to the starting position and appearance and disappearance dynamics in experiments or according to the simulated flake distributions (see above). |  |
| Agents starting positions | According to the starting positions in experiments or in the center of the arena for simulated flake distributions. |  |

| Model types and names | Description (see *Figure 3—figure supplement 1C*) |
|---|---|
| Independent model (IND model) | fish only respond to flakes within $D_f$ around them ($D_n$=0) |
| Attraction to consumption events (Att_feed) | fish respond to flakes within $D_f$, otherwise respond to neighbors' previous flake detections within $D_n$ |
| Attraction to neighbors regardless of consumption (Att model) | fish respond to flakes within $D_f$, otherwise attract to the center of mass (average position of the group) of their neighbors within $D_n$, regardless of consumption events |
| Alignment with neighbors (Align model) | fish respond to flakes within $D_f$, otherwise align with t average direction of neighbors within $D_n$ |
| Attraction to consumption events and alignment (Att_feed+Align model) | fish respond to flakes within $D_f$, otherwise respond to neighbors' previous flake detections within $D_n$, otherwise align with neighbors within $D_n$ |

*Continued on next page*

*Continued*

| Model types and names | Description (see *Figure 3—figure supplement 1C*) |
|---|---|
| Attraction and alignment with neighbors (Att+Align) | fish respond to flakes within $D_f$, otherwise align with neighbors within $D_n/2$, and attract to the center of mass of neighbors within $D_n/2 < d_n < D_n$ regardless of consumption events. If neighbors exist in both zones, fish average the attraction and alignment responses. |
| Attraction to consumption events and attraction and alignment with neighbors (Att$_{feed}$+Att+Align model) | fish respond to flakes within $D_f$, otherwise respond to neighbors' previous flake detections within $D_n$, otherwise align with neighbors within $D_n/2$, and attract to the center of mass of neighbors within $D_n/2 < d_n < D_n$ regardless of consumption events. If neighbors exist in both zones, fish average the attraction and alignment responses. |

## Model fitting

Models were fitted to data by finding the parameter values ($D_f$ for the independent model, and $D_f, D_n$ for the social models) that maximized the log-likelihood of the simulated consumption times given the empirical consumption times of a given group (see below) and minimized the normalized squared error between the swimming statistics (average polarity and nearest neighbor distance) of real groups and simulated groups.

Consumption times: we used the distribution of consumption times of flake i over the simulations (100 repetitions), to assess the probability of observing a sequence of consumption events on real data traces. Thus, the probability of the i-th consumption event to occur at time $T$ is given by its value from the simulations of the model, $P_{D_f,D_n}(T_{data}(i))$, and the probability of a sequence of consumption times is given by the product of the probabilities of the individual events $\prod_i P_{D_f,D_n}(T_{data}(i))$.

The log-likelihood of the model is then given by:

$$logP_{D_f,D_n}(T_{data}(1), T_{data}(2), ...) = \sum_i logP_{D_f,D_n}(T_{data}(i))$$

where $T_{data}(i)$ is the actual consumption time of the $i^{th}$ flake, and the probability $P$ is the distribution of consumption times of the $i^{th}$ flake over all simulations for a specific set of model parameters $D_f, D_n$. We used kernel density smoothing to estimate a continuous probability from the discrete distribution obtained from simulations.

Polarity and nearest neighbor distance: we calculated the normalized mean squared error between the statistics obtained from simulations and the ones observed in the data: $E(D_f, D_n, S) = \frac{(\langle \hat{s} \rangle - \langle s \rangle)^2}{SD(\hat{s})}$ where $S$ is the relevant statistic measured (polarity or nearest neighbor distance), $\hat{S}$ is the value obtained from simulations and <...> denotes the average. polarity is defined $Polarity = \frac{1}{N}\sum_i \vec{d}_i$, which is the average direction vectors of the N fish in the group.

Combined accuracy measure: To assess the accuracy of the model we combined these three separate measures into a single error function to be minimized by searching over the $D_f, D_n$ values of the simulations: $E^{combined} = E^{consumptions} + E^{polarity} + E^{D_{nn}}$ where $E^{consumptions}$ was taken as minus the log likelihood of consumptions (see above), and all three error surfaces were standardized by subtracting their mean value and dividing by the standard deviation, such that all errors will have similar units.

## Measuring flake clustering

To quantify the spatial clustering of flakes, we simulated the random positions of N flake consumptions (corresponding to the number of actual flake consumptions of each of the real groups). We then calculated the average nearest neighbor distance of the simulated flakes $D_{nn1}^{rand}$ and repeated this analysis 100,000 times to obtain a distribution of average nearest neighbor distances expected at random. We then compared the actual average nearest neighbor distance $D_{nn1}$ of flake consumptions to the random distribution and assessed how likely it is to obtain $D_{nn1}$ if flakes were randomly distributed in space. The level of clustering of a group of n flakes is then defined by: $\hat{D}(n) = \frac{nn1(n) - \mu_{D_{nn1}^{rand}}(n)}{\sigma_{D_{nn1}^{rand}}(n)}$, where $\mu_{D_{nn1}^{rand}}$ and $\sigma_{D_{nn1}^{rand}}$ are the mean and standard deviation of the distribution of nearest neighbor distances of randomly distributed flakes.

## Inequality measure

To quantify the inequality of flake consumptions between members of the same group, we calculated the *Theil* index of inequality (*Theil, 1967*). This is an information theory-based measure, assessing the difference in the entropy of a distribution from the maximum entropy expected if consumption rate was equal for all agents, and is given by

$$I_{Theil}(k,n) = \frac{1}{k}\sum_{i=1}^{k} \frac{n_i}{\mu} log\left(\frac{n_i}{\mu}\right)$$

where $k$ is the number of agents, $n_i$ is the number flakes consumed by fish $i$ and $\mu$ is the mean number of flakes consumed by a fish in the group. In addition, we normalized $I_{Theil}$ by $log\,k$, the maximum possible value if one fish consumed all flakes. We then quantified equality by $1 - \frac{I_{Theil}}{log\,k}$, where 1 indicates full equality and 0 is full inequality.

## Sample sizes and power estimation

As the current research tests novel effects of social behavior on group foraging, precise estimation of sample sizes and statistical power could not be conducted a priori. Instead, we have based our choice of sample sizes on previously published studies of collective behavior and social foraging behavior of zebrafish (*Miller and Gerlai, 2007*; *Harpaz et al., 2017*). In addition, we chose to include more than one group size in the study design (groups of three and six fish) to support the generality of our findings. Finally, as the study is focused on groups, sample sizes were chosen to be large enough to conduct parametric and nonparametric statistical testing (e.g. Pearson's correlation coefficient, Wilcoxon's rank-sum test), while minimizing the total number of animals used in the experiments.

## Acknowledgements

We thank Udi Karpas, Ori Maoz, Oren Forkosh, Tal Tamir, Adam Haber, Hanna Zwaka, Gasper Tkacik, Yadin Dudai, Ofer Feinerman, and Iain Couzin for discussions and suggestions. This work was supported by a HFSP grant, European Research Council Grant 311238, a Israel Science Foundation Grant 1629/12, and a Simons Foundation grant, as well as research support from Martin Kushner Schnur and Mr. and Mrs. Lawrence Feis. ES is the incumbent of the Joseph and Bessie Feinberg Professorial Chair.

## Additional information

### Funding

| Funder | Grant reference number | Author |
| --- | --- | --- |
| Israel Science Foundation | 1629/12 | Elad Schneidman |
| European Research Council | 311238 | Elad Schneidman |
| Human Frontier Science Program | RGP0065/2012 | Elad Schneidman |

| Simons Foundation | 542997 | Elad Schneidman |

The funders had no role in study design, data collection and interpretation, or the decision to submit the work for publication.

## Author contributions

Roy Harpaz, Conceptualization, Software, Formal analysis, Validation, Visualization, Methodology, Writing - original draft, Writing - review and editing; Elad Schneidman, Conceptualization, Software, Formal analysis, Supervision, Funding acquisition, Validation, Visualization, Methodology, Writing - original draft, Project administration, Writing - review and editing

## Author ORCIDs

Roy Harpaz (ID) https://orcid.org/0000-0001-9587-3389
Elad Schneidman (ID) https://orcid.org/0000-0001-8653-9848

## Ethics

Animal experimentation: Animal care and all the experimental procedures were approved by the Institutional Animal Care and Use Committee of the Weizmann Institute of Science (Protocol 17310415-2).

## Decision letter and Author response

Decision letter https://doi.org/10.7554/eLife.56196.sa1
Author response https://doi.org/10.7554/eLife.56196.sa2

## Additional files

### Supplementary files

• Transparent reporting form

### Data availability

All data used in this work have been made available via the main author's public GitHub account: https://github.com/schneidmanlab/zebrafishForaging (copy archived at https://github.com/elifes-ciences-publications/zebrafishForaging).

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
