## [Decision Letter]

**Acceptance summary:**

This submission presents a study of social foraging in zebrafish that combines empirical data from foraging groups in the lab with simulation models describing their behavior. The main results are that: (1) a model incorporating an attraction of fish to neighbors who have recently eaten a food item improves model fits to empirical data; (2) simulating social foraging interactions leads to increased foraging efficiency for both individuals and groups; and (3) adding social interactions in the simulations leads to more equal food distributions. This work is an example of the power of developing careful and data-driven modeling of animal behavior, and of using such models to link the actions of individual animals to generate insight into collective responses.

**Decision letter after peer review:**

Thank you for submitting your article "Social interactions drive efficient foraging and income equality in groups of fish" for consideration by *eLife*. Your article has been reviewed by three peer reviewers, and the evaluation has been overseen by a Reviewing Editor and Christian Rutz as the Senior Editor. The following individual involved in the review of your submission has agreed to reveal their identity: Orit Peleg (Reviewer #2).

The reviewers have discussed their reviews with one another, and the Reviewing Editor has drafted this decision letter to help you prepare a revised submission. We would like to draw your attention to changes in our revision policy that we have made in response to COVID-19 (https://elifesciences.org/articles/57162).

Summary:

This submission presents a study of social foraging in zebrafish that combines empirical data from foraging groups in the lab with spatially explicit simulation models describing their behavior. The main results are that: (1) a model incorporating an attraction of fish to neighbors who have recently eaten a food item improves model fits to the empirical data over a model assuming independent foraging; (2) simulating such social foraging interactions leads to increased foraging efficiency at both individual and group levels both for the empirical distributions of food and for a range of other distributions; and (3) incurring social interactions in the simulations also leads to a greater equality of food distribution.

Revisions:

Although all of the reviewers thought the work was timely and of relevance to a larger audience, nicely tying together empirical data and computational modeling, they agreed that some additional work is necessary to better support key claims, particularly with regard to the specific choice of model and how the specific choice of model relates to the conclusions. We ask the authors to address both the revisions here, as well as the minor points listed further below.

1) The reviewers' main concern was that the social model is just one out of the many possible models. What in the data suggests that this model could in principle be better than others? The question is pertinent, because the authors make many suggestions that results of further analysis of the model is relevant to the understanding of fish collective behavior. One possibility would be to simulate the model we suggested in comment 1 below; we do not ask the authors to prove the negative that no other model could possibly produce the results seen in the data, but we do ask that they frame their results accordingly, either through additional simulations or added discussion of the literature.

2) The authors should clarify how the initial positions of the fish (related to the position "small start box"), relative to the initial positions of the food sources were chosen. The reviewers were not able to find it in the main text or in the Materials and methods. This is important, because variations in the initial distance to a food source could affect the first passage time of the first detection, either by individuals or by groups.

3) Another concern is that, for groups of 3 fish, one would expect to have aggressive encounters as the fish are in a small group, they are hungry, and there is food present. Is this common effect observed in this set-up? How does it affect the results?

Other comments:

1. A major suggestion is to consider including an additional simulation model to serve as a comparison for both the asocial model and the social model. In particular, we see two issues with the model comparison as it currently stands. First, there is the very simple issue that a model with more parameters will necessarily yield a better fit to the data. As the analysis stands currently, the social model has another parameter (D_n) compared to the asocial model, and as only likelihood is used this ignores model complexity. The second issue is that we think a null model should be developed to help distinguish between the effects of social interactions per se vs. the effects of attraction to others in the context of food detection. There has already been some evidence in fish schools that gradient following can emerge through simple interactions in which no explicit gradient following and/or attraction to those who find food is included (see e.g. Berdahl et al. 2013 and Torney et al. 2009). At the moment, these two mechanisms are confounded in the social model. Both of these issues could be addressed by the inclusion of a single comparison model in which social attraction is included (perhaps at random times rather than when food is found) but not explicitly linked to food. This would allow the authors to address the question of whether the fish are really responding to their neighbours finding food, or whether they are simply just attracted to their neighbours overall. We suspect this comparison model will be a worse fit to the data, however it would be important to include it to rule out this possibility. In addition, it would make for an interesting comparison of how fish perform in terms of foraging efficiency when they explicitly move towards those who found food vs. just exhibit a general schooling tendency. We suspect that adding in this additional model would not require too much additional work, since it amounts to only a small change to the simulated rules of the fish

---

## [Author Response]

Revisions:Although all of the reviewers thought the work was timely and of relevance to a larger audience, nicely tying together empirical data and computational modeling, they agreed that some additional work is necessary to better support key claims, particularly with regard to the specific choice of model and how the specific choice of model relates to the conclusions. We ask the authors to address both the revisions here, as well as the minor points listed further below.1) The reviewers' main concern was that the social model is just one out of the many possible models. What in the data suggests that this model could in principle be better than others? The question is pertinent, because the authors make many suggestions that results of further analysis of the model is relevant to the understanding of fish collective behavior. One possibility would be to simulate the model we suggested in minor point #1; we do not ask the authors to prove the negative that no other model could possibly produce the results seen in the data, but we do ask that they frame their results accordingly, either through additional simulations or added discussion of the literature.

We have made substantial changes to the manuscript following this major point, which reinforce and clarify the original conclusions. Specifically, following the reviewer’s request, we simulated and analyzed additional models of fish schooling and shoaling, which are based on models of collective behavior in fish groups that were previously suggested in the literature. We now present a comparison of 7 different models – one in which fish were acting independently of one another, and 6 different variants of social interactions between fish. These social models rely on three kinds of interactions between fish: attraction to other fish, alignment with other fish, attraction to neighbors’ previous feeding events, and combinations of these three. This comparison now allows us to better delineate the role of these interactions in group foraging.

The analysis of this expanded set of models conforms with and strengthens the conclusions from the first version of the manuscript: We again show the key role of attraction to neighbors’ feeding events. The most accurate model was the one that relied on attraction to feeding events in combination with "alignment force” between fish. This model captured the feeding rates of the groups as well as other swimming statistics, namely nearest neighbor distance and group polarity. We show that the attraction to feeding locations of neighbors is the key component that predicts the increase in feeding rates in groups and the increase in equality of flake distribution, compared to independent foragers. The social models that do not incorporate this behavior showed no increase or a decrease in foraging efficiency and equality within groups.

Notably, this was a major change to the detailed analysis and the text of the manuscript. Consequently, we changed and expanded Figures 3 and 4 that describe and compare the models, and updated Figures 5, 6 and 7, and the corresponding supplementary figures Figure 3—figure supplement 1, Figure 4—figure supplement 1, Figure 5—figure supplement 1, Figure 6—figure supplement 1, Figure 6—figure supplement 2, Figure 7—figure supplement 1, Figure 7—figure supplement 2 and Videos 3-8.

2) The authors should clarify how the initial positions of the fish (related to the position "small start box"), relative to the initial positions of the food sources were chosen. The reviewers were not able to find it in the main text or in the Materials and methods. This is important, because variations in the initial distance to a food source could affect the first passage time of the first detection, either by individuals or by groups.

Flakes were randomly distributed in a ring of 36cm<R<95cm around the start box, which was positioned in the middle of the arena (text was revised to clarify this, lines 591-595). Because the first passage of time to a flake may indeed vary, due to individual dispersion patterns of the flakes and the initial velocities and headings of the fish, we used the rate b_k_ of flake consumption from the fit to the consumption times given by the equation log T^k(n)=nbk+ak , since it is not affected by the first passage of time to a flake consumption. For comparison, we also replot Figure 1B and 1C where time zero is the consumption of the first flake, and show that results are similar (see Author response image 1).

**Author response image 1. sa2fig1:** 

3) Another concern is that, for groups of 3 fish, one would expect to have aggressive encounters as the fish are in a small group, they are hungry, and there is food present. Is this common effect observed in this set-up? How does it affect the results?

Indeed, zebrafish were previously shown to monopolize clumped food sources in laboratory experiments, and also might show aggressive behaviors while foraging (e.g. Grant and Kramer, 1992, Animal Behavior). Consequently, we constructed our experiments to reduce these effects: 1. Fish were housed in their designated groups for at least a month prior to experiments to allow for social hierarchies to become well established (Miller et al., 2017, Journal of Neuroscience). 2. Flakes were scattered across the arena and no specific location could be associated with being rich or poor in terms of food abundance, to reduce food monopolization. In addition, to test for aggressive encounters that might affect fish foraging, we compared the number of close contacts between fish that were accompanied by a sharp increase in speed – as these would imply an aggressive encounter took place – before foraging (i.e. during habituation without food) and during foraging (with food). We did not observe an increase in the number of close contacts when fish were foraging for food in any of the group sizes tested; in groups of 6 fish the number of close contacts was actually reduced during foraging – as shown in Author response image 2

Other comments:1. A major suggestion is to consider including an additional simulation model to serve as a comparison for both the asocial model and the social model. In particular, we see two issues with the model comparison as it currently stands. First, there is the very simple issue that a model with more parameters will necessarily yield a better fit to the data. As the analysis stands currently, the social model has another parameter (D_n) compared to the asocial model, and as only likelihood is used this ignores model complexity. The second issue is that we think a null model should be developed to help distinguish between the effects of social interactions per se vs. the effects of attraction to others in the context of food detection. There has already been some evidence in fish schools that gradient following can emerge through simple interactions in which no explicit gradient following and/or attraction to those who find food is included (see e.g. Berdahl et al. 2013 and Torney et al. 2009). At the moment, these two mechanisms are confounded in the social model. Both of these issues could be addressed by the inclusion of a single comparison model in which social attraction is included (perhaps at random times rather than when food is found) but not explicitly linked to food. This would allow the authors to address the question of whether the fish are really responding to their neighbours finding food, or whether they are simply just attracted to their neighbours overall. We suspect this comparison model will be a worse fit to the data, however it would be important to include it to rule out this possibility. In addition, it would make for an interesting comparison of how fish perform in terms of foraging efficiency when they explicitly move towards those who found food vs. just exhibit a general schooling tendency. We suspect that adding in this additional model would not require too much additional work, since it amounts to only a small change to the simulated rules of the fish.

We followed the reviewer’s suggestion and as we explain above in the response to the first major concern, we believe that the new set of models that we studied addresses these issues in detail. Specifically, we now show that models that only rely on general attraction to other fish or alignment forces between fish give a poor fit to the data, and do not exhibit an increase in foraging efficiency and income equality compared to independent foragers. Only models that had a specific attraction to neighbors’ feeding showed these traits. We emphasize that all the models we compared had similar complexity in terms of the number of free parameters.